# Experimental infection of *Artibeus lituratus* bats and no detection of Zika virus in neotropical bats from French Guiana, Peru, and Costa Rica suggests a limited role of bats in Zika transmission

**Alvaro Aguilar-Setién**[1]*, **Mónica Salas-Rojas**[1], **Guillermo Gálvez-Romero**[1], **Cenia Almazán-Marín**[1], **Andrés Moreira-Soto**[2], **Jorge Alfonso-Toledo**[1], **Cirani Obregón-Morales**[1], **Martha García-Flores**[1], **Anahí García-Baltazar**[1], **Jordi Serra-Cobo**[3,4], **Marc López-Roig**[3,4], **Nora Reyes-Puma**[5], **Marta Piche-Ovares**[6,7], **Mario Romero-Vega**[6], **Daniel Felipe Barrantes Murillo**[8¤], **Claudio Soto-Garita**[6], **Alejandro Alfaro-Alarcón**[8], **Eugenia Corrales-Aguilar**[6], **Osvaldo López-Díaz**[9], **Dominique Pontier**[10], **Ondine Filippi-Codaccioni**[10], **Jean-Baptiste Pons**[10], **Jeanne Duhayer**[10], **Jan Felix Drexler**[2]

1 Instituto Mexicano del Seguro Social, Coordinación de Investigación Médica, Unidad de Investigación en Inmunología. Hospital de Pediatría, Mexico City , México, 2 Institute of Virology, Charité –Universitätsmedizin Berlin, Helmut-Ruska-Haus, Berlin, Germany, 3 Departament de Biologia Evolutiva, Ecologia i Ciències Ambientals Facultat de Biologia, Universitat de Barcelona, Barcelona, Spain, 4 Institut de Recerca de la Biodiversitat (IRBIO). Facultat de Biolia. Universitat de Barcelona, Barcelona, Spain, 5 Instituto de Medicina Tropical "Daniel Alcides Carrión" Universidad Nacional Mayor de San Marcos, Lima, Peru, 6 Virology-CIET (Research Center for Tropical Disease), University of Costa Rica, San José, Costa Rica, 7 Department of Virology, School of Veterinary Medicine, National University, Heredia, Costa Rica, 8 Department of Pathology, School of Veterinary Medicine, National University, Heredia, Costa Rica, 9 Departamento de Producción Agrícola y Animal, Universidad Autónoma Metropolitana Unidad Xochimilco, Mexico City, Mexico, 10 Université de Lyon, Université Lyon 1, CNRS, Laboratoire de Biométrie et Biologie Evolutive UMR5558, Villeurbanne, France

¤ Current address: Department of Pathobiology, College of Veterinary Medicine, Auburn University, Auburn, Alabama, United States of America
* balantiopterix@gmail.com

## Abstract

Bats are important natural reservoir hosts of a diverse range of viruses that can be transmitted to humans and have been suggested to play an important role in the Zika virus (ZIKV) transmission cycle. However, the exact role of these animals as reservoirs for flaviviruses is still controversial. To further expand our understanding of the role of bats in the ZIKV transmission cycle in Latin America, we carried out an experimental infection in wild-caught *Artibeus lituratus* bats and sampled several free-living neotropical bats across three countries of the region. Experimental ZIKV infection was performed in wild-caught adult bats (4 females and 5 males). The most relevant findings were hemorrhages in the bladder, stomach and patagium. Significant histological findings included inflammatory infiltrate consisting of a predominance of neutrophils and lymphocytes, in addition to degeneration in the reproductive tract of males and females. This suggests that bat reproduction might be at some level affected by ZIKV. Leukopenia was also observed in some inoculated animals.

**Data Availability Statement:** All relevant data are within the manuscript and its Supporting Information files.

**Funding:** This work was supported the German Federal Foreign office (https://www.bundesregierung.de/breg-en/federal-government/ministries/federal-foreign-office) by the GLACIER Global Centre for Health and Pandemic Prevention from the German academic exchange services (DAAD) (Grant agreement: 57592642) (https://www.daad.de/en/information-services-for-higher-education-institutions/further-information-on-daad-programmes/glacier/), the German Centre for Infection Research (DZIF) through the ZIKApath project and the European Union's Horizon 2020 research and innovation program through the ZIKAlliance project (grant agreement 734548) (JFD). The work in Mexico was partially supported by CONACYT – FONCICYT (https://conahcyt.mx) Project "Una Alianza Global para Controlar y Prevenir el Virus del Zika" Number 274386 (AAS). The work in Costa Rica was also supported by the FEES-CONARE (Fondo Especial para la Educación Superior- Consejo Nacional de Rectores) B7362 project (https://www.conare.ac.cr/) (AMS). The funders had no role in study design, data collection and analysis, decision to publish, or preparation of the manuscript.

**Competing interests:** The authors have declared that no competing interests exist.

Hemorrhages, genital alterations, and leukopenia are suggested to be caused by ZIKV; however, since these were wild-caught bats, we cannot exclude other agents. Detection of ZIKV by qPCR was observed at low concentrations in only two urine samples in two inoculated animals. All other animals and tissues tested were negative. Finally, no virus-neutralizing antibodies were found in any animal. To determine ZIKV infection in nature, the blood of a total of 2056 bats was sampled for ZIKV detection by qPCR. Most of the sampled individuals belonged to the genus *Pteronotus* sp. (23%), followed by the species *Carollia* sp. (17%), *Anoura* sp. (14%), and *Molossus* sp. (13.7%). No sample of any tested species was positive for ZIKV by qPCR. These results together suggest that bats are not efficient amplifiers or reservoirs of ZIKV and may not have an important role in ZIKV transmission dynamics.

## Author summary

In previous works in 2008–2009, we found the presence of antibodies against flaviviruses, and viral RNA was detected in Neotropical chiropterans in Mexico, which led us to support the hypothesis that these animals could be reservoirs of flaviviruses. As controversial opinions have been exposed and based on a previous (2019) experimental ZIKV infection experiment conducted at Colorado State University using adult *Artibeus* males from a captive colony, in this work, we also experimentally infected adult *Artibeus* males complementarily adding females and using wild-caught animals instead of laboratory bats. We also monitored a diverse range of natural bat populations in Latin America for the presence of viral RNA against ZIKV in blood. A plaque reduction seroneutralization test was used for the detection of antibodies against ZIKV. Similar to the previous work, we found histopathological alterations in male testicles but also in the ovaries and oviducts of females, as well as gliosis and multifocal necrosis in pyramidal neurons and Purkinge cells of inoculated animals. Only two urine samples from inoculated animals showed viral RNA. Additionally, leukopenia and lymphoid follicular splenic hyperplasia were evidenced. In contrast to what was reported, no neutralizing antibodies against ZIKV were detected in any sample. Viral RNA within the blood was not present in any of the 2056 bat samples collected in French Guiana, Peru and Costa Rica and proceeding from 34 bat genera. These results together suggest that bats are not efficient amplifiers or reservoirs of ZIKV and might not have an important role in ZIKV transmission dynamics.

## Introduction

Zika virus (ZIKV) belongs to the genus *Flavivirus* of the *Flaviviridae* family [1]. ZIKV was first isolated in Africa in 1954 [1,2]. The virus circulates between an urban transmission cycle involving arthropod vectors and humans and a sylvatic cycle involving arthropod vectors and nonhuman primates [2,3]. In addition, ZIKV alternative transmission routes in humans include sexual, blood transfusion and perinatal routes [4–7]. ZIKV was first reported in the Americas in 2015, and since then, it has spread throughout the continent, causing more than 850,000 human cases [8]. During an outbreak, serological surveys performed in Brazil detected a high 60% population exposure [8]. This high population exposure suggested the end of the outbreak due to high herd immunity. However, if ZIKV adapts to new vertebrate wildlife hosts

in the Americas, as yellow fever virus (YFV) has [9], ZIKV might establish a sylvatic cycle until there is a large enough naïve population to cause another outbreak.

Early during the outbreak, modeling studies suggested nonhuman primates as the most likely ZIKV hosts [10]. However, serological surveys among wild nonhuman primates in Brazil found a limited role in ZIKV transmission and urged the study of other wildlife species as possible ZIKV hosts [11,12]. Considering the richness of vertebrate species in America, the role of nonprimate animals in the ZIKV cycle remains understudied [13]. Bats are a megadiverse group of mammals only outnumbered by rodent diversity on the American continent. Molecular and serological detection of flaviviruses suggests exposure to flaviviruses in the wild [14–19]. For the most studied flavivirus in bats, dengue virus (DENV), a peridomestic study found only limited exposure of bats, likely due to proximity to humans and consumption of DENV vectors [20]. Additionally, *in vitro* studies in which several neotropical bat cell lines were infected to search for serological and molecular evidence of infection in wild bats by experimentally infecting *Artibeus* bats with DENV showed that these species are inadequate DENV hosts and may not play an important role in DENV transmission [21–24]. For ZIKV, experimental infections and field studies performed in the 1950s-1960s documented the probable susceptibility and disease development in African and American bat species [25–27]. Recently, an experimental infection in a breeding colony of *Artibeus jamaicensis* bats detected ZIKV RNA and seroconversion in some studied animals and raised the possibility that bats may have a role in Zika virus ecology that may even endanger bat populations [28]. Considering the neotropical bat species richness and limited information of ZIKV hosts to date, we carried out an experimental infection in wild-caught *Artibeus* bats and sampled free-living neotropical bats across several Latin American countries to assess their role in the ZIKV transmission cycle in the Americas.

## Materials and methods

### Ethics statement

Capture and animal handling were performed according to Mexican environmental standards (permit SEMARNAT Mexico No SGPA/DGVS/08986/18).

This study and its associated protocols were designed based on national ethical legislative rules and approved by the Institutional Committee of Care and Use of Animals of the University of Costa Rica (CICUA-042-17), Committee of Biodiversity of the University of Costa Rica (VI-2994-2017), National System of Conservation Areas (SINAC): Tempisque Conservation Area (Oficio-ACT-PIM-070-17), and La Amistad-Caribe Conservation Area (M-PC-SI-NAC-PNI-ACLAC-047-2018).

Authorization for bat capture in French Guiana was provided by the Ministry of Ecology, Environment, and Sustainable Development during 2015–2020 (approval no. C692660703) from the Departmental Direction of Population Protection (DDPP, Rhône, France). All methods (capture and animal handling) were approved by the Muséum National d'Histoire Naturelle, Société Française pour l'Étude et la Protection des Mammifères, and the Direction de l'Environnement, de l'Aménagement et du Logement (DEAL), Guyane. Bat sampling in Peru was authorized by the Research Ethics Committee of the Institute of Tropical Medicine "Daniel Alcides Carrion" (Constancia de aprobación CIEI-2017-16). The survey did not involve endangered or protected species.

### Capture of bats for infection experiments

Eleven great fruit-eating bats (*Artibeus lituratus*) were captured in Oaxtepec, Morelos, Mexico (18°54′23″N 98°58′13″W) in May 2018. Animals were measured, weighed, aged, sexed, tagged,

and identified as *Artibeus lituratus* species using a dichotomous key by field specialists. Capture and animal handling were performed according to Mexican environmental standards (permit SEMARNAT Mexico No SGPA/DGVS/08986/18). The captured bats were individually placed in cotton bags, hydrated using an electrolyte solution and transported to the animal facility in Mexico City, where they were housed in groups of 4 and 3 individuals in 45 x 45 x 80 cm cages. Bats were quarantined for 1 month before the experiment.

The first human cases of ZIKV in Morelos, Mexico were reported in December 2015, reaching over 267 confirmed cases between 2015 and 2017; subsequently, human cases progressively decreased in the area (PAHO) [29]. To ensure that the animals were not acutely infected by ZIKV, urine, blood and plasma samples were taken four times during the quarantine period and tested with ZIKV-specific qRT–PCR according to a previously described methodology [30]. Plasma samples taken in the quarantine period were submitted to a virus neutralization test by plaque reduction neutralization test as described below [31]. Nine bats were selected for the experimental infection, and two were selected as controls. Animals were kept in captivity with food and water *ad libitum*, according to the Guidelines of the American Society of Mammalogists for the Use of Wild Mammals in Research [32].

## Experimental infection

**Virus stock.**   A ZIKV isolate from a human patient in Yucatan, Mexico (ZIKV/Mer. IPN01) in 2017 was used. Briefly, C6/36 cells previously grown with 12 ml of Eagle's minimal essential medium (DMEM) with 10% fetal bovine serum (FBS) were inoculated with ZIKV at a multiplicity of infection (MOI) of 0.1 and incubated in DMEM without FBS at 27˚C for 1 hour, after which new medium with FBS was added. The infected cells were kept until a cytopathic effect was observed in >80% of the cells. Next, the cell supernatant was collected, centrifuged, and stored in aliquots at −70˚C. The viral stock titers were determined in BHK-21 cells using standard plaque assays in carboxymethylcellulose (CMC).

**Experimental infection.**   Nine bats were injected subcutaneously with $2\times10^5$ plaque-forming units (PFUs) of Zika virus and two control bats with FBS in the scapular area on day 0 (**Fig 1A**). On day -1, blood samples were taken for ZIKV serology and for complete blood cell count as described below. To observe ZIKV pathogenesis over time, bats were divided into 4 groups by sex: group 1 consisted of female MO04 and male MO06 euthanized 3 days postinoculation (d.p.i.), group 2 consisted of male MO01 and female MO08 euthanized 7 d.p.i., group 3 consisted of males MO05 and MO02 and female MO09 euthanized 14 d.p.i., and group 4 consisted of male MO07 and female MO10 euthanized 21 d.p.i. Euthanasia of the control group of female MO11 and male MO12 was performed 21 d.p.i. (**Fig 1A**).

Bats were observed daily, and body temperature and weight were recorded. Behavioral changes or signs suggestive of diseases such as lethargy, nasal or oral discharge, urine color, and pasty and/or diarrheic feces consistency were recorded. Oral swabs and urine (placing each animal in individual containers for 30 minutes) were collected daily (**Fig 1A**). On the day of euthanasia, the bats were anesthetized using 30 μl of a mixture of 1:1 of 100 mg/mL ketamine–100 mg/mL xylazine and euthanized by cardiac exsanguination. Next, blood samples were divided for molecular and serological ZIKV evaluation, and whole blood was used for complete blood count (CBC). The analysis of the blood to obtain the CBC was carried out in the VETSCAN HM5 Hematology Analyzer-Zoetis previously calibrated for Phyllostomidae bat leucocytes using the referenced previous reports (33–36). The samples for which no result was obtained in any parameter after repeating the test were analyzed manually. The differential count of leukocytes was carried out based on blood films stained with Giemsa stain for the visualization of the cellular morphological characteristics in the smear. Different organs, such

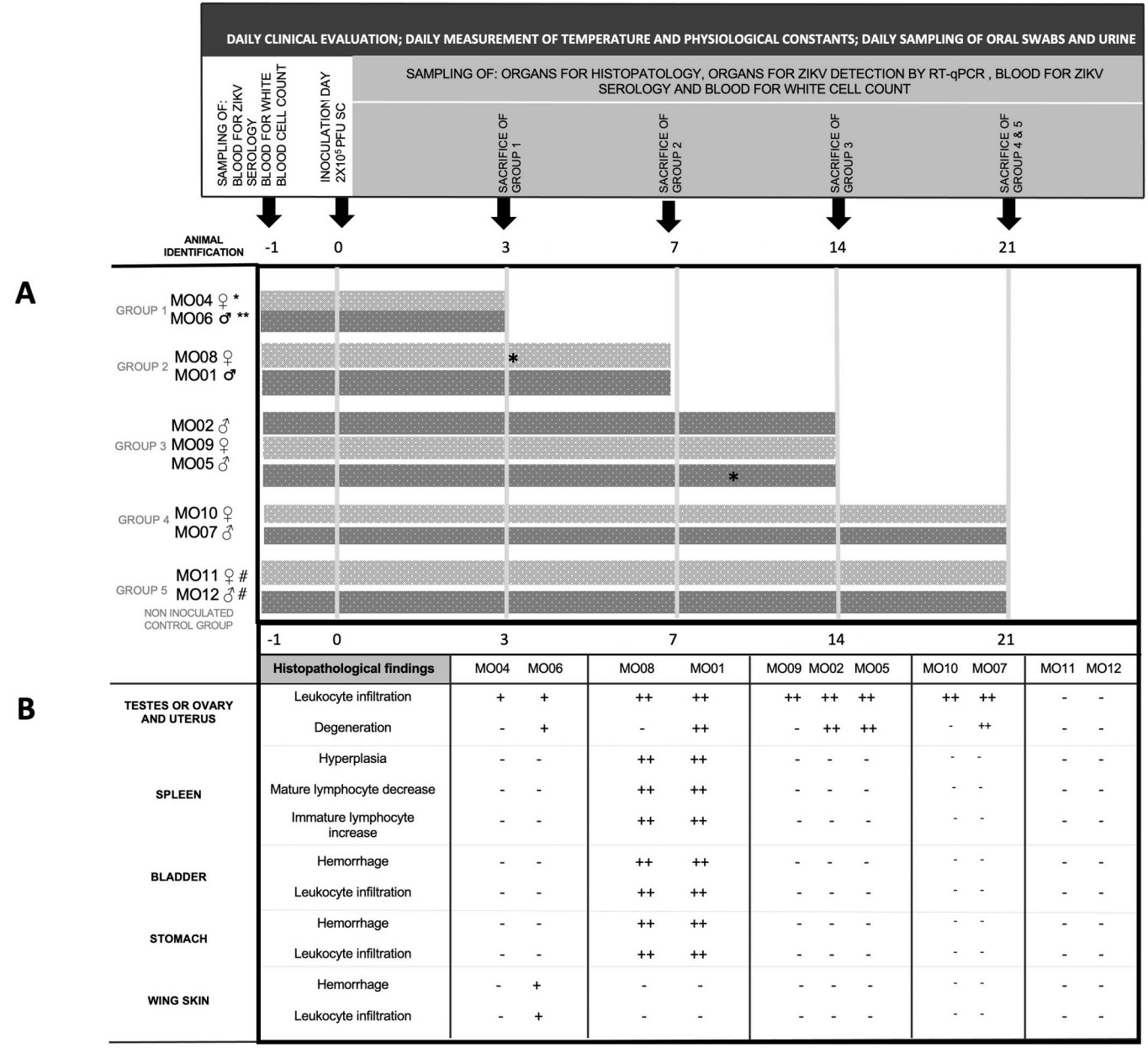

**Fig 1. Experimental design and timeline.** Bats were inoculated (SC) with 2 X 105 PFUs of ZIKV Mer.IPN01. Groups, sampling, and time of sacrifice A. Histopathological and gross findings B.- No changes observed.+ Mild.++ Moderate.* Urine samples positive for ZIKV RNA by qRT-PCR. Only two urine samples tested positive for ZIKV by RT–qPCR, ranging from 3.19 X 102 3.61 X 102 copies/mL, urine sample from bat MO08 on day 3 p.i. and urine samples from bat MO05 on day 9 p.i. All other samples (urine, swabs, and tissues) tested negative.# control group.

as the brain, tongue, salivary glands, heart, lung, liver, spleen, gut, kidney, bladder, uterus, ovary, and testicles, were collected on the day of euthanasia from each group. Each sampled organ was divided into two parts. A part of each organ was stored at −80˚C until processed for ZIKV RNA detection by qRT–PCR. The remaining part of each organ was fixed in 10% buffered formalin and routinely processed for histopathological evaluation.

## Laboratory analyses

**Histological studies.**   The formalin and paraformaldehyde-fixed tissues were processed using standard methodology: paraffin embedded, 5-μm thick tissues were sectioned and stained with Hematoxylin and Eosin (H&E). Slides were analyzed by two veterinary pathologists who observed the slides without any contact with each other and using a blinded approach as part of routine pathological work.

**Plaque reduction neutralization test.**   A plaque reduction neutralization test (PRNT) using ZIKV strain H/PF/2013 was performed. Briefly, sera were diluted 1:10, 1:40, 1:100 and 1:350 in serum-free DMEM. Subsequently, a mixture of 35 μL of each serum dilution plus 35 μL of ZIKV containing forty plaque-forming units was incubated for 1 hour at 37˚C with 5% $CO_2$. After this time, 50 μL of the serum dilution and ZIKV mixture plus 250 μL of cell culture media was added to previously seeded Vero cells and incubated for 1 hour at 37˚C with 5% $CO_2$. After the incubation, the cell media was changed, and the cells were incubated for 4 days and fixed with 6% paraformaldehyde. Plaques were visualized using a violet crystal solution. A positive result was measured when a serum dilution reached < 50% of the number of plaques counted in the controls [31].

**Real-Time RT–PCR.**   RNA extraction for the experimental infection and sampling was carried out using the MagNA Pure 96 DNA and Viral NA Small Volume Kit in the MagNA pure 96 system (Roche, Germany). The presence of ZIKV was detected using ZIKV-specific real-time PCR primers (Bonn- NS1 and Bonn-E), as described previously [30].

## Fieldwork

**A sampling of bats in different countries of Latin America.**   All sampling and capture were performed with the proper wildlife permits and ethics clearances of the host countries. Samples were gathered before and after the ZIKV epidemics in the Americas between 2010 and 2019 in Peru, Costa Rica, and French Guiana. The complete data, including sites, species and years where and when bat samples were obtained, are included in **Fig 2** and **Tables 1 and S3**. According to the reports of the Pan American Health Organization (PAHO) [29], Zika cases among humans were presented as follows: Peru.- In epidemiological week (EW) 17 of 2016, Peru notified the PAHO [29] of the detection of the first case of autochthonous vector-borne transmission of Zika to EW 37 of 2016, and low numbers of Zika cases were reported. From EW 38 onward, the number of cases began to increase, reaching more than 800 cases registered in EW 14 of 2017. Since then, cases have progressively declined. Costa Rica.- In EW 4 of 2016, the detection of the first autochthonous vector-borne transmission of Zika was reported in Costa Rica. Since the emergence of Zika, weekly numbers of suspected and confirmed cases increased steadily up to EW 36 of 2016, after which a downward trend was observed. During 2017, transmission continued, and a new increase in the incidence of Zika cases was observed between EW 19 and 33 of 2017. Since then, cases have progressively declined. French Guiana.- In EW 51 of 2015, French Guiana notified the PAHO [29] of the detection of the first case of autochthonous vector-borne transmission of Zika, reaching between 100 and 600 suspected cases each EW from January to August 2016, which subsequently fell to less than 50 cases in 2017. Considering these dates, the samples taken before the Zika pandemic were 368 and corresponded entirely to samples taken in French Guiana; the samples taken at the beginning or during the pandemic were a total of 1688 and were taken in the three sampled countries (Peru, Costa Rica and French Guiana) (**Table 1**). Bats were captured using mist nets in neotropical forests, degraded forests, urban sites, and caves, and no specific bat species were targeted. After capture, the bats were kept individually in cotton bags until examination. Trained specialists identified the species using the morphological keys for

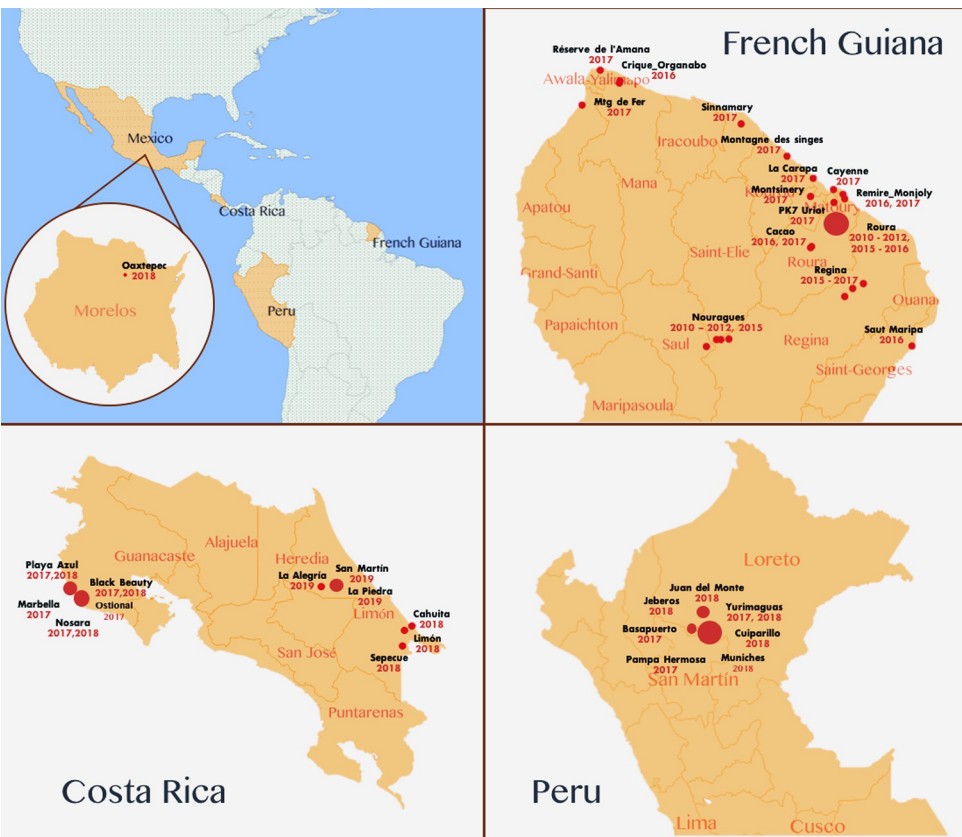

**Fig 2. Sites and years where and when bat samples were obtained.** Note.- The size of the dots is proportional to the number of years sampled.

the species. Blood samples for ZIKV detection by qPCR were taken by a trained veterinarian or biologist either from the brachial vein in case the animal was being marked and released, which was the case for bats in Peru and French Guiana, or intracardiac puncture after intramuscular application of euthanasia for bats in Costa Rica (corresponding permits in ethical statements). The blood sample was immediately centrifuged, and the serum was stored at 4°C until it arrived at the laboratory, where it was stored at -80°C until molecular analyses were performed.

## Results

### Experimental infection

**Clinical signs of ZIKV disease in bats.**   None of the captured bats were ZIKV positive through qPCR before the experiment. For the nine adult *Artibeus lituratus* bats and the two controls, no clinical signs suggestive of disease were recorded during the quarantine period. The inoculated bats showed a mean temperature (geometric mean: 35.6°C; standard deviation (SD) 0.75; 95% confidence interval (CI)) similar to that of the noninfected control bats (geometric mean: 35.7°C; SD 0.6; 95% CI). Two inoculated bats showed extreme temperatures (male MO07 on day 3 p.i., 38.5°C and male MO02 on day 6 p.i., 38.3°C) (**S1 Fig and S1 Table**); this temperature was not at odds with the mean temperature for Phyllostomidae bats, which ranges between 34.5 and 39.5°C [33,34]. Other studies also showed no temperature

**Table 1. List of bats sampled in French Guiana, Peru, and Costa Rica from 2010 to 2017 for the detection of ZIKV in blood by RT–PCR and antibodies by seroneutralization test.**

| Species | Guiana | | | | | | Peru | | Costa Rica | | | Total by species |
|---|---|---|---|---|---|---|---|---|---|---|---|---|
| | 2010 | 2011 | 2012 | 2015 | 2016 | 2017 | 2017 | 2018 | 2017 | 2018 | 2019 | |
| *Anoura caudifer* | | | | | 1 | | 9 | 22 | | | | 32 |
| *A. geoffroyi* | 48 | 29 | 74 | 40 | 50 | 13 | | | | | | 254 |
| *Artibeus sp.* | 1 | | | | | 1 | | | | | | 2 |
| *A. jamaicensis* | | | | | | | | | 12 | 9 | | 21 |
| *A. lituratus* | | | | | 3 | 1 | 2 | | | | | 6 |
| *A. obscurus* | | | | | 2 | 8 | | 8 | | | | 18 |
| *A. phaeotis* | | | | | | | | | | 3 | | 3 |
| *A. planirostris* | | | | | 21 | 35 | | 2 | | | | 58 |
| *Carollia sp.* | | | | | | | 5 | | | | | 5 |
| *C. brevicauda* | | | | | 1 | | | | | | | 1 |
| *C. castanea* | | | | | | | | | | 6 | | 6 |
| *C. perspicillata* | 3 | 6 | 14 | 6 | 97 | 119 | 37 | 42 | | 7 | 4 | 335 |
| *Chiroderma sp.* | | | | | 1 | | | | | | | 1 |
| *C. trinitatum* | | | | | | | | | 1 | | | 1 |
| *Cynomops planirostris* | | | | | | 4 | | | | | | 4 |
| *Dermanura cinerea* | | | | | 9 | 16 | | | | | | 25 |
| *D. gnoma* | | | | | | 1 | | | | | | 1 |
| *Desmodus rotundus* | 1 | 1 | 5 | 13 | | 2 | | 18 | | | | 40 |
| *Eumops auripendulus* | | | | | 2 | | | | | | | 2 |
| *Eptesicus chiriquinus* | | | | 1 | | 1 | | | | | | 2 |
| *Glossophaga commissarisi* | | | | | | | | | | | 1 | 1 |
| *G. soricina* | | | | | | | | | 4 | 2 | 11 | 17 |
| *Linchonycteris obscura* | | | | | | | | | 1 | | | 1 |
| *Lionycteris spurrelli* | | | | | 1 | | | | | | | 1 |
| *Lonchophylla thomasi* | | | 1 | | | | | | | | | 1 |
| *Lonchorhina inusitata* | | | | | 3 | 3 | | | | | | 6 |
| *Lophostoma brasiliense* | | | | | | 1 | | | | | | 1 |
| *L. silvicola* | | | | | | 2 | | | | | | 2 |
| *Mimon crenulatum* | | | | | | 1 | | | | | | 1 |
| *Molossus sp* | | | | | | 1 | | | | | | 1 |
| *M. barnesi* | | | | | | 1 | | | | | | 1 |
| *M. molossus* | | | | | 56 | 35 | 88 | 88 | | | 1 | 268 |
| *M. paranus* | | | | | 2 | | | | | | | 2 |
| *M. rufus* | | | | | 9 | 1 | | | | | | 10 |
| *Micronycteris microtis* | | | | | | | | | | 1 | | 1 |
| *Myotis nigricans* | | | | | | | 65 | 111 | | 1 | 5 | 182 |
| *Noctilio albiventris* | | | | | 2 | 3 | | | | | 1 | 6 |
| *N. leporinus* | | | | | 1 | 25 | | | | | | 26 |
| *Phylloderma stenops* | | | | | 2 | 1 | | | | | | 3 |
| *Phyllostomus discolor* | | | | | | 1 | | | 1 | | | 2 |
| *P. latifolius* | 5 | 1 | 2 | 1 | | | | | | | | 9 |
| *P. elongatus* | | | | 1 | | | 2 | | | | | 3 |
| *P. hastatus* | | | | 20 | 16 | 30 | 5 | 4 | | | | 75 |
| *Platyrrhinus albericoi* | | | | | | | | 4 | | | | 4 |
| *P. brachycephalus* | | | | | 3 | 8 | | | | | | 11 |

*(Continued)*

**Table 1.** (Continued)

| Species | Guiana | | | | | | Peru | | Costa Rica | | | Total by species |
|---|---|---|---|---|---|---|---|---|---|---|---|---|
| | 2010 | 2011 | 2012 | 2015 | 2016 | 2017 | 2017 | 2018 | 2017 | 2018 | 2019 | |
| *P. fusciventris* | | | | | | 9 | | | | | | 9 |
| *P. helleri* | | | | | 1 | | | | | 1 | | 2 |
| *P. incarum* | | | | | 3 | 1 | | | | | | 4 |
| *P. infuscus* | | | | | | | 14 | 7 | | | | 21 |
| *Pteronotus sp.* | 61 | 79 | 33 | **119** | 85 | 81 | | | | | | 458 |
| *P. gymnonotus* | | | | | | 11 | | | | | | 11 |
| *P. parnellii* | | | | | | | | | | | 1 | 1 |
| *Peropteryx macrotis* | | | | | | | 7 | | | | | 7 |
| *Rhinophylla pumilio* | | | | | 1 | | | | | | | 1 |
| *Rhogeessa io* | | | | | | | | | | 1 | | 1 |
| *Rhynchonycteris naso* | | | | | | | 1 | | | | 2 | 3 |
| *Saccopteryx leptura* | | | | | | | | | 1 | | | 1 |
| *Sturnira lilium* | | | | | 6 | 20 | | | 5 | 1 | | 32 |
| *S. tildae* | | | | | 17 | 8 | | | | | | 25 |
| *Tonatia saurophila* | 1 | | | | | 3 | | | | | | 4 |
| *Trachops cirrhosus* | 2 | 1 | | | 6 | 2 | | | 1 | | | 12 |
| *Uroderma bilobatum* | | | | | 2 | 4 | | | | | | 6 |
| *U. convexum* | | | | | | | | | | 5 | | 5 |
| *Vampyressa tyone* | | | | | | | | 1 | | | | 1 |
| Total | 122 | 117 | 129 | **201** | 403 | 453 | 235 | 307 | 26 | 37 | 26 | 2056 |

Total samples before Zika pandemic: 368
Total samples after Zika pandemic: 1688

alterations when bats were experimentally infected with other viruses, such as coronavirus [35,36] and Marburg virus [37].

**Complete blood count.** The red blood cell count showed that several individuals from the different groups, sampled at different times (from day 0 to day 21) throughout the experiment, had some hemoglobin deficiencies, i.e., all individuals showed moderately low levels of MHC (12–14 pg) (**Table 2**). In addition, clear normochromic microcytic anemia was found in MO02 on day 14 pi, and MO09 presented marginal erythrocytosis on day 14 p.i. The most relevant results in the white blood cell count compared to the values given for *Artibeus* bats (4.7 ± 2.4 X 10$^9$/mL) (32–34) showed leukocytosis with lymphocytosis and monocytosis in individuals MO06 and MO08, in addition to lymphocytosis in MO09 and leukocytosis with lymphocytosis in MO11 (**Tables 2 and S2**). The animals MOO4 and MO06 sacrificed on day 3 and animals MO01 and MO08 sacrificed on day 7 showed clear leukopenia with neutropenia. Individuals MO02 and MO09 sacrificed on day 14 pi presented neutropenia with lymphocytosis, whereas MO09 showed a decrease in neutrophils with respect to day 0. MO05, MO07, MO10 and MO11 only presented lymphocytosis.

**Gross findings.** During the necropsy, the main macroscopic lesions were observed in three bats; MO06 presented few petechial hemorrhages in the patagium 3 days p.i. (**Fig 3**), MO01 and MO08 presented mild to moderate petechial hemorrhages in the patagium and stomach mucosa, and MO08 presented moderate petechial hemorrhages in the bladder mucosa that caused hematuria at 7 d.p.i. No further apparent pathological changes were observed in the necropsy; the macroscopic lesions found in infected bats suggested a possible

**Table 2. Complete blood count.**

| Parameters | Reference values | Post-infection days | | | | | | | | | | | | |
|---|---|---|---|---|---|---|---|---|---|---|---|---|---|---|
| | | 0 | | | 3 | | 7 | | 14 | | | 21 | | |
| | | MO06 | MO08 | MO09 | MO04 | MO06 | MO01 | MO08 | MO05 | MO02 | MO09 | MO07 | MO10 | MO11[#] |
| RBC $10^{12}$/L | 10.0–13.0 | 11.79 | 11.79 | 12.36 | IS | IS | IS | IS | 11.54 | 9.53[a] | 11.43 | 11.76 | 12.33 | 13.09 |
| HGB g/L | 155–195 | 156 | 154[a] | 174 | IS | IS | IS | IS | 153[a] | 132[a] | 139[a] | 148[a] | 155 | 159 |
| HCT L/L | 0.50–0.65 | 0.525 | 0.599 | 0.598 | IS | IS | IS | IS | 0.485[a] | 0.397[a] | 0.535 | 0.518 | 0.567 | 0.591 |
| MCV fL | 46.5–53.5 | 44.5[a] | 50.8 | 48.4 | IS | IS | IS | IS | 42.0[a] | 41.7[a] | 46.8 | 44.0[a] | 46.0[a] | 45.1[a] |
| MHC pg | 14.3–16.3 | 13.2[a] | 13.1[a] | 14.1[a] | IS | IS | IS | IS | 13.3[a] | 13.9[a] | 12.2[a] | 12.6[a] | 12.6[a] | 12.1[a] |
| MCHC g/L | 290–322 | 297.1 | 257.1[a] | 291 | IS | IS | IS | IS | 315.5 | 332.5 | 259.8[a] | 285.7[a] | 273.4[a] | 269[a] |
| WBC $10^9$/L | 2.3–7.1 | 7.62[b] | 8.26[b] | 6.88 | 0.7*[a] | 0.3*[a] | 0.5*[a] | 0.9*[a] | 3.64 | 5.78 | 2.58 | 5.43 | 6.50 | 8.13[b] |
| Neutrophils | 1599.35–5420.81 | 3536 | 3643 | 3347 | 287[a] | 114[a] | 265[a] | 144[a] | 1762 | 1422[a] | 289[a] | 1695 | 2873 | 4057 |
| Lymphocytes | 95.05–1570.01 | 3239[b] | 3866[b] | 3227[b] | 350 | 162 | 230 | 702 | 1594[b] | 4312[b] | 2118[b] | 3394[b] | 3471[b] | 3845[b] |
| Monocytes | 20.91–528.47 | 846[b] | 752[b] | 269 | 35 | 24 | 5[a] | 54 | 284 | 46 | 173 | 386 | 156 | 228 |
| Neutrophils % | 32.5–74.1 | 46.4 | 44.1 | 48.8 | 42.0* | 38.0* | 53.0* | 16.0*[a] | 48.4 | 24.6[a] | 11.2[a] | 30.4[a] | 44.2 | 49.9 |
| Lymphocytes % | 2.0–33.6 | 42.5[b] | 46.8[b] | 46.9[b] | 50.0*[b] | 54.0*[b] | 46.0*[b] | 78.0*[b] | 43.8[b] | 74.6[b] | 82.1[b] | 62.5[b] | 53.4[b] | 47.3[b] |
| Monocytes % | 0.4–11.1 | 11.1 | 9.1 | 4.3 | 5.0* | 8.0* | 1.0* | 6.0* | 7.8 | 0.8 | 6.7 | 7.1 | 2.4 | 2.8 |

* Determination by blood smear assessment.

IS, insufficient sample.

[a] values below the reference.

[b] values above the reference.

[#] control group.

MO, bat number.

cause of ZIKV infection; however, since these were bats captured in the wild, we could not exclude the presence of other agents, such as filariasis [38] or DENV infection, that cause bleeding in the skin and mucous membranes during experimental infections in *Artibeus* bats [23]. In summary, because these lesions can occur in various physiopathogeneses, more data are needed to consider their characteristic findings in ZIKV infections.

**Histopathology.** The males infected in the four groups demonstrated histological changes within the epididymis and testicle described as follows. The epididymal epithelium was markedly vacuolated, with loss of stereocilia (**Fig 4A and 4B**). Mild interstitial lymphocyte infiltrates and mild to moderate germ cell degeneration were observed in the epididymis. Epididymal ducts were filled by degenerate to apoptotic spermatids. In the testicle, the seminiferous tubules had early stages of testicular degeneration, with marked vacuolization of the Sertoli cells, with evidence of apoptotic spermatids (**Fig 4C and 4D**) in addition to mild congestion and few to moderate neutrophils and macrophages infiltrating the interstitium.

Similarly, all infected females presented mild to moderate lymphocytic infiltrates within the ovary and uterus throughout all the days after inoculation. The oviduct presented a minimal neutrophilic infiltrate within the lamina propria and degeneration of secretory and epithelial cells that progressively increased until day 21 without showing regression or lymphocytic exocytosis. The uterine lumen was filled by a moderate amount of a granular eosinophilic proteinaceous secretion intermixed with a mild neutrophilic exudate (**Fig 4E and 4F**).

The most representative lesions observed in the brain of bat MO08 sacrificed 7 days after infection were present in the hippocampus, which showed areas with moderate multifocal necrosis in pyramidal neurons, in addition to moderate multifocal gliosis areas. Multifocal necrosis of the Purkinje cells was present within the cerebellum (**Fig 5A and 5B**). However, it

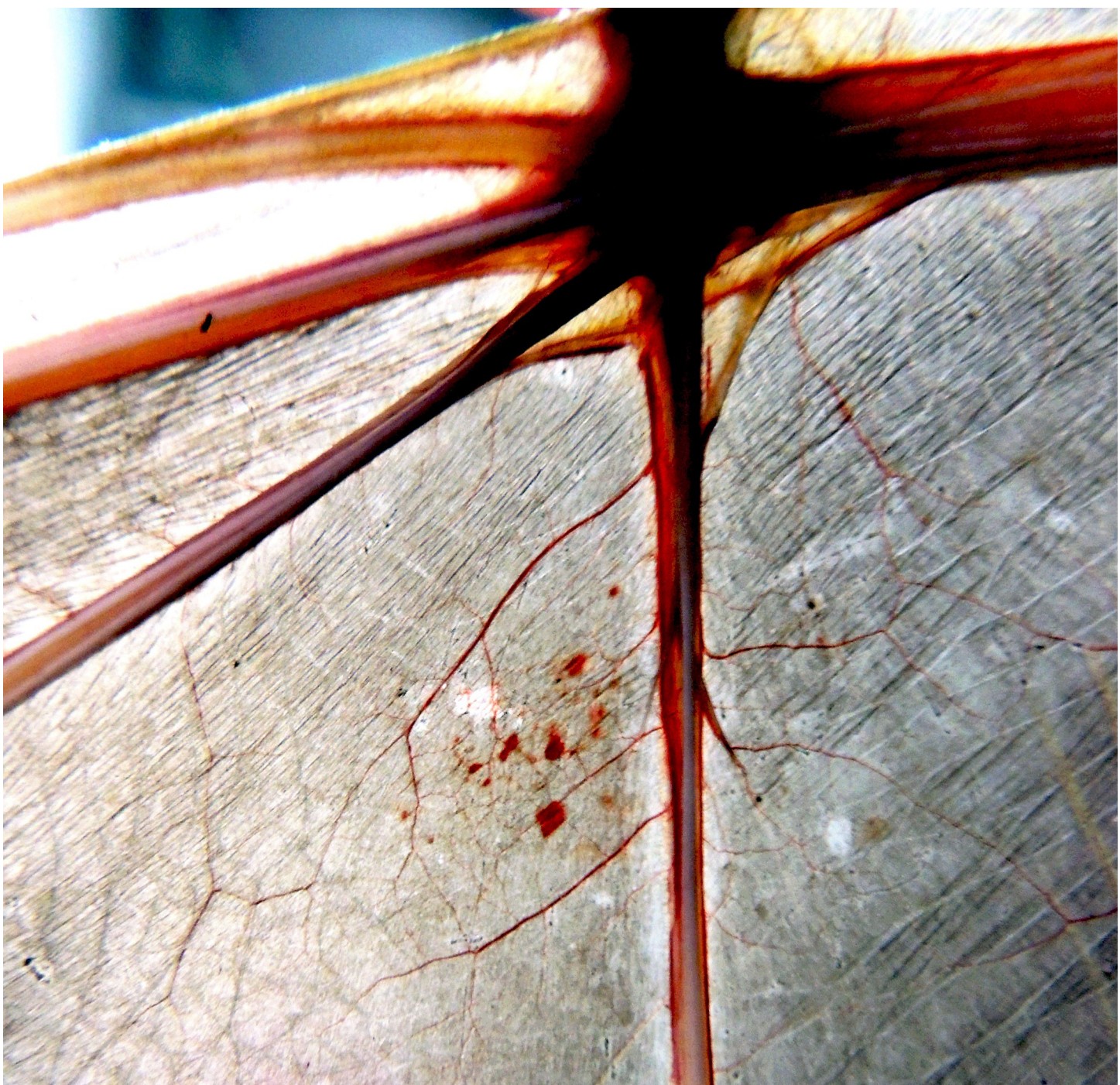

**Fig 3. Petechiae on the right wing of bat MO06 observed 3 days p.i.**

should also be considered that these findings could be due to other pathogens or parasites since these animals were wild caught.

The bats MO06, MO01 and MO08 presented moderate dermal hemorrhages in the patagium. The dermis was moderately infiltrated by neutrophils and lymphocytes (**Fig 6**). Dermal capillaries were markedly distended and congested with neutrophilic leukostasis. Finally, the

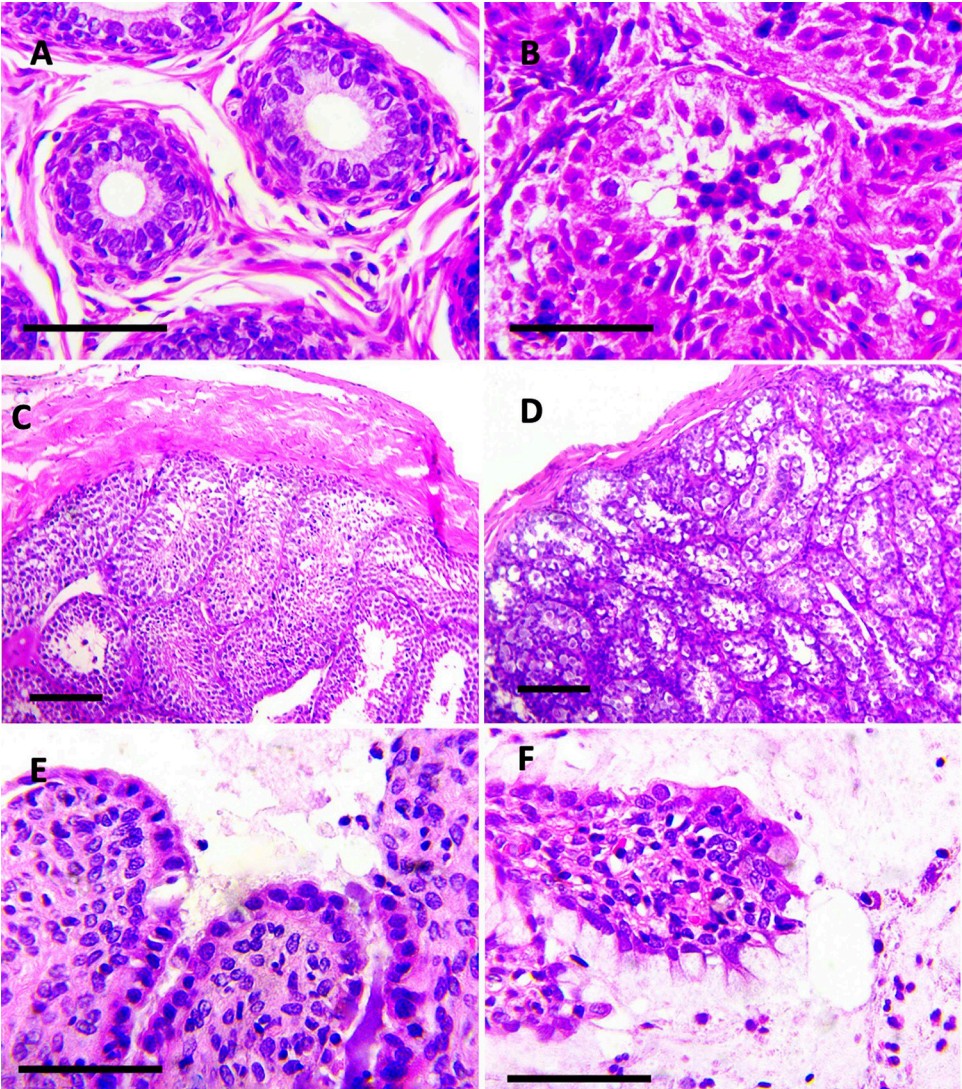

**Fig 4. Photomicrographs of sections of the epididymis.** A. Control epididymal ducts. H&E stain; bar = 50 μm. B. Epididymal duct epithelial degeneration, vacuolization, and loss of stereocilia. The lumen is filled by abundant degenerate to apoptotic spermatids. H&E stain; bar = 50 μm. Photomicrographs of sections of the testicles. C. Control testicular parenchyma. D. Early stage of testicular degeneration, vacuolated Sertoli cells with disorganized exfoliated or absent germ cells. H&E stain; bar = 50 μm. Photomicrographs of sections of the oviduct. E. The lamina propria is infiltrated by scant neutrophils. F. The degenerated epithelium has lymphocytic exocytosis, and the oviduct lumen is filled by granular eosinophilic proteinaceous secretion intermixed with neutrophils. H&E stain; bar = 50 μm.

spleens of MO01 and MO08, which were sacrificed 7 days pi (Group 2), showed moderate lymphoid depletion and follicular lymphoid hyperplasia (**Fig 7**). This observation was consistent with the WBC of these individuals who presented leukopenia due to neutropenia, and additionally, the total number of lymphocytes decreased drastically compared to the individuals sampled on day 0, including MO08, which was sampled on day 0 and day 7 (**Tables 2 and S2**). This finding is compatible with what has been reported during multiple viral infections that generate hypo-proliferation or lympholysis.

**Limited detection of viral RNA in urine.**   Consistent with Malmlov et al. 2019 [28], who detected ZIKV RNA in low copy numbers ($1x10^2$-$1x10^3$ copies/ml) in the brain and urine of

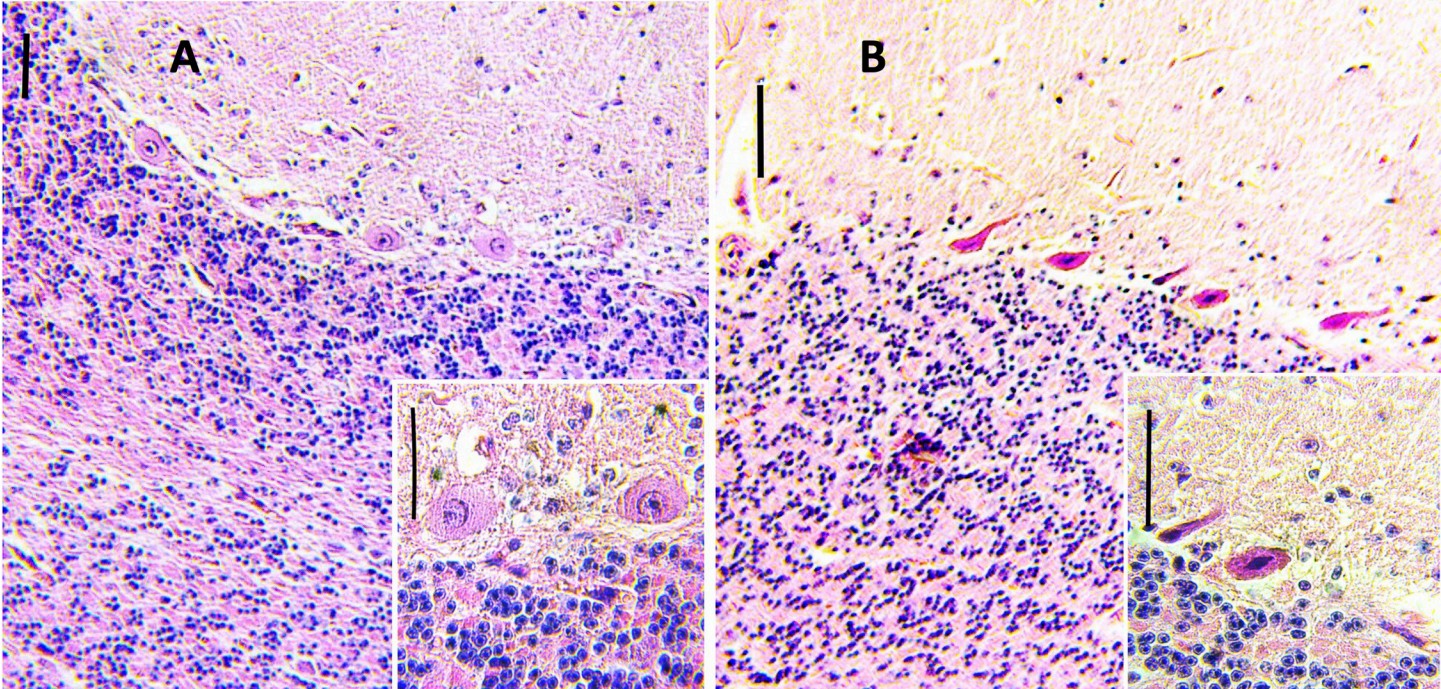

**Fig 5. Photomicrographs of sections of the cerebellum.** A. Control: molecular, Purkinje cell, and granular layer. H&E stain; bar = 50 μm. Purkinje cells *(inset)*. H&E stain; bar = 50 μm. B. infected: Photomicrographs of sections of the cerebellum. Necrosis of the Purkinje cell layer. H&E stain; bar = 100 μm. Necrotic cells are shrunken, with angulated cellular margins, hypereosinophilic cytoplasm, and pyknotic nuclei *(inset)*. H&E stain; bar = 50 μm.

three individual bats, we detected ZIKV in only two urine samples of male individuals (20% detection rate, CI: 4.5–52.0%) in urine from bat MO08 at 3 d.p.i. and MO05 at 9 d.p.i., ranging from $3.19 \times 10^2$- $3.61 \times 10^2$ copies/ml (Table 1). ZIKV RNA could not be detected in any other tissue or oral swab.

**Seroneutralization test.** On the experimental infection, no serum had a neutralizing effect on the ZIKV strain H/PF/2013, in contrast with the findings of Malmlov et al. 2019 [28], who reported modest seroconversion by an enzyme-linked immunosorbent assay (ELISA) test (3200 vs. >12000 in convalescent human sera) in three bats on day 28 pi. Putative reasons for the contrasting findings in our bats include the higher specificity of PRNT versus ELISA. On the other hand no Ab against ZIKV was detected in 2056 individual blood serum samples from bats of 34 genera collected in 3 Latin American countries (Peru, French Guiana, and Costa Rica) before (368 samples) and during (1688 samples) ZIKV pandemics (2010–2019) (Table 1). These results are consistent with the negative detection of ZIKV antibodies by PRNT in 7 African bat species [39]. However, these results are in contrast with the frequent detection of ZIKV antibodies in free-ranging bats in early African studies [25,27]. A major weakness of these early studies is the possible lack of differentiation among flaviviruses due to the testing method: the hemagglutination inhibition test (HIT).

## Latin American-wide bat sampling

**No ZIKV was detected in free-living bats in different countries of Latin America.** A total of 2056 bats were sampled (blood for ZIKV detection by qRT–PCR) belonging to different neotropical species and countries (Peru, French Guiana, and Costa Rica). In French Guiana, samples were collected after and during ZIKV pandemics in the Americas over the years

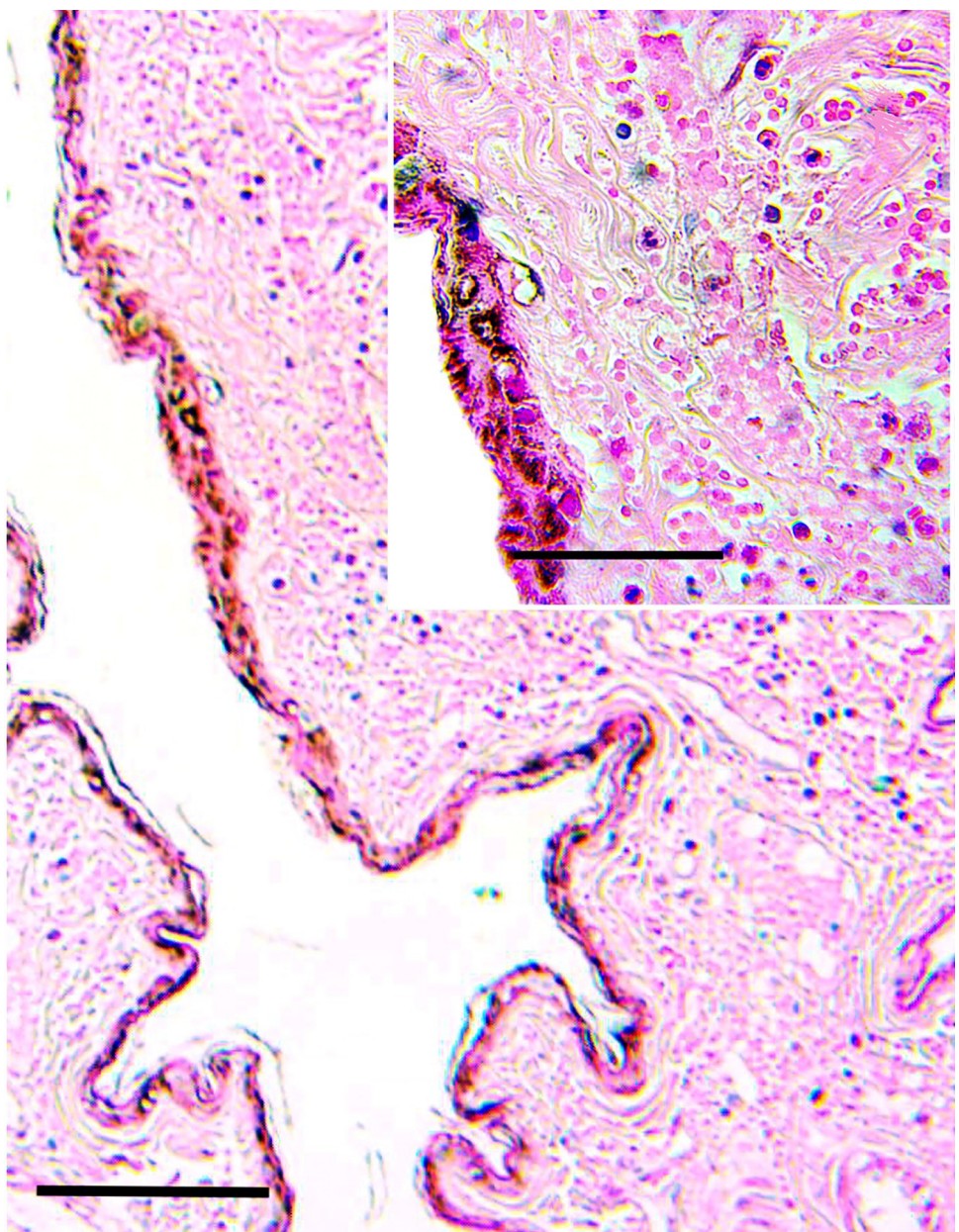

**Fig 6. Photomicrographs of sections of the patagium.** H&E stain; bar = 200 μm. The dermis has hemorrhages and has a mild interstitial neutrophilic infiltrate (*inset*). H&E stain; bar = 50 μm.

2010, 2011, 2012, 2015, 2016, and 2017; in Costa Rica during ZIKV pandemics over 2017, 2018, and 2019; and in Peru, samples were collected during ZIKV pandemics only over 2017 and 2018 (**Fig 2** and **Table 1**). Sampling included 6 bat families and 34 genera, of which 22 were family Phyllostomidae (*Anoura, Artibeus, Carollia, Chiroderma, Dermanura, Desmodus, Glossophaga, Linchonycteris, Lionycteris, Lonchophylla, Lonchorhina, Lophostoma, Mimon, Micronycteris, Phylloderma, Phyllostomus, Platyrrhinus, Rhinophylla, Sturnira, Tonatia, Trachops,* and *Uroderma*); 3 were family Molossidae (*Cynomops, Eumops* and *Molossus*); 3 were Vespertilionidae (*Eptesicus, Myotis* and *Rhogeessa*); 3 were family Emballonuridae (*Peropteryx,*

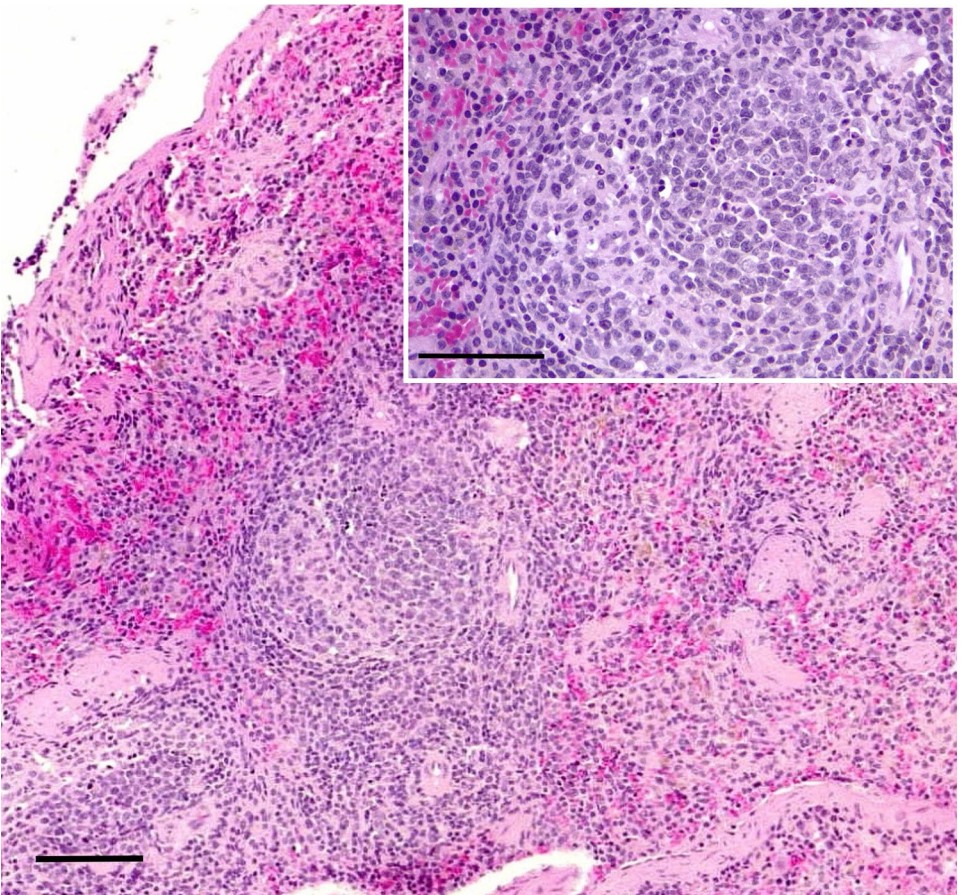

**Fig 7. Photomicrographs of sections of the spleen.** The white pulp has marked lymphoid follicular hyperplasia. H&E stain; bar = 100 μm. Lymphoid follicles have prominent germinal centers. H&E stain; bar = 50 μm.

*Rhynchonycteris*, and *Saccopteryx*); one was family Noctilionidae (*Noctilio*), and one was family Mormoopidae (*Pteronotus*); within these families, 60 species were identified (**Table 1**). The largest number of sampled individuals belonged to the genus *Pteronotus* sp. (n = 470, 23%), followed by the species *Carollia* sp. (n = 347, 17%); *Anoura* sp. (n = 286, 14%), and *Molossus* sp. (n = 282 13.7%) (**Table 1**). No blood sample was positive for ZIKV by qPCR. This lack of detection is consistent with a serosurvey of Brazilian and African bats that also reported a lack of detection [39,40]. However, a sampling bias cannot be excluded due to the diversity of bat species in Latin America. Urine samples were not analyzed, and it must be considered that some studies in humans show that ZIKV RNA can be detected at higher levels and for a longer time after the onset of infection in the urine compared to blood and other fluids [30,41,42].

## Discussion

In this study, neither viral RNA nor Ab of ZIKV was detected in 2056 individual blood serum samples from bats of 34 genera collected in 3 Latin American countries (Peru, French Guiana, and Costa Rica) before (368 samples) and during (1688 samples) ZIKV pandemics (2010–2019) (**Table 1**). Moreover, scarce viral RNA and no seroconversion were found in ZIKV-infected *Artibeus* bats. However, some pathological changes were observed in the

experimentally infected animals (**Fig 1**). Taken together, these results allow us to assume that neotropical bats from different genera might not have an important role in ZIKV transmission dynamics.

Our findings from 9 captured *Artibeus lituratus* bats from both sexes confirmed and expanded upon Malmlov et al.'s [28] experimental infection findings. Only two out of 180 (1%) urine samples taken from ZIKV-inoculated animals showed low ZIKV copy numbers by qPCR. All other urine samples, swabs and tissues from inoculated animals were negative. These results, together with the lack of seroconversion found, suggest that *Artibeus* bats are not efficient amplifiers or reservoirs of ZIKV. The same seems to be true for DENV as well, since Cabrera et al. in 2014 reported that experimental inoculation of *Artibeus jamaicensis* Bats with DENV serotypes 1 or 4 showed no evidence of sustained replication, and our group in 2013 reported limited replication of DENV 2 in these same animals [22,23]. In Malmlov's work [28], only 3 positive samples by PCR were reported, two of which were urine samples and the third was nervous tissue, while in the present work, only two samples were positive, and both were urine samples. In addition, some studies in humans show that ZIKV RNA can be detected at higher levels and for a longer time after the onset of infection in urine compared to blood and other samples [41,42]; thus, urine may be used as a preferred sample for further studies aiming to detect ZIKV in wild bats.

The histopathological findings were consistent with experimental infections in male *Artibeus* bats [28]. In addition, the results were reminiscent of experimental murine ZIKV infections, which reported severe infection with inflammatory infiltration and degeneration of reproductive tissues and cells, but in these cases, infections were performed by the genital route in mice infected during the diestrus-like phase [43] or by the parenteral route in transgenic or treated mice lacking components of the innate antiviral response [44,45].

The genital alterations observed suggest that bat reproduction might be affected to some extent by ZIKV.

Further studies analyzing whether the degeneration observed leads to infertility or alterations during pregnancy will be needed to assess whether ZIKV could cause a disruption in bat reproductive health. Nevertheless, in the Americas, Zika has been closely associated with *Aedes* genus mosquitoes [2], especially *Ae. aegypti*, which are highly anthropophilic. Experimentally, *Artibeus* bats were bitten by *Aedes aegypti* mosquitoes in an attempt to infect these mammals with DENV (22), suggesting that *Aedes* mosquitoes could feed on bats. However, it is not known how often this happens. DENV and ZIKV have been isolated from mosquitoes other than *Aedes aegypti* that could be part of the sylvatic cycle of those flaviviruses [2]. However, it is not known whether these mosquito species frequently feed on bats or are consumed by bats [46]. In addition, it should be considered that the competence of these vectors depends on the level and duration of viremia in a given host, as was shown by Fernandes et al. (2020) for *Aedes aegypti* and rhesus macaques [47].

Between days 3 and 7 after infection, clear leukopenia was observed in infected animals with subsequent recovery after day 9 pi (**Table 2**). Generally, in mammals, as the first line of defense against infectious processes, the immune reaction includes leukocytosis in association with fever, and during viral infections, leukopenia and subsequent lymphocytosis can also occur, as occurred in our research. However, in some lipopolysaccharide (LPS)-stimulated insectivores, no leukocytosis or fever was observed [48]. Another feature of leukocyte kinetics reported in Vespertilionidae bats in Karelia is the clear leukopenia generated during hibernation in bats [49]. These authors concluded that the differences in the proportion of lymphocytes may be related to the biological, ecological, and physiological peculiarities of the bats studied, which shows the need for further studies to determine a correlation between bat infection with ZIKV and leukocyte kinetics of *Artibeus*.

Finding some clinical alterations with scarce viral RNA detection could be somewhat contradictory, and one explanation could be the hypersensitivity phenomena produced by certain viral proteins. In 2015, Modhiran et al. reported an analogy between the cellular biology of bacterial lipopolysaccharides (LPS) and that of DENV nonstructural protein 1 (NS1) [50]. LPS and DENV NS1 interact with Toll-like receptor 4 (TLR 4) on the surface of monocytes, macrophages, and endothelial cells, inducing the release of a range of cytokines and chemokines. These same cytokines circulate in the blood of patients with hemorrhagic Dengue syndrome, producing hemorrhages and tissue destruction [51]. However, we do not know if this could be extrapolated to ZIKV; therefore, more research in this regard is necessary to better understand the immune system of bats against viral diseases. It would be interesting to prove this hypothesis by the inoculation of inactivated ZIKV or its components in our animal model. On the other hand, a recent report in Brazil, confirmed the presence of ZIKV MR766 (African lineage), in free-living neotropical non-human primates (*Alouatta guariba*)[52]. So far, ZIKV MR766, has only been isolated in non-human primates. Therefore, the ecology of different lineages of ZIKV in neotropical wildlife is yet to be clarified.

Finally it is necessary to point out that in this work, the sampled individuals of certain species of wild bats were scarce (only 1 or 2 sampled individuals). This was due to several factors such as the density of their populations, the accessibility of their habitat, their behavior, etc. The relative significance of these negative results for these species is obvious. In this case, only positive results would be significant.

## Supporting information

**S1 Fig. Corporal temperature recorded in experimentally infected bats during the experiment.**
(TIF)

**S1 Table. Supplementary data for corporal temperature recorded in experimentally infected bats during the experiment.**
(XLSX)

**S2 Table. Supplementary data for the complete blood count.**
(XLSX)

**S3 Table. Supplementary data for sites (geographical coordinates) and years where bat samples were obtained.**
(XLSX)

## Acknowledgments

Zika virus ZIKV/Mer. IPN01, was kindly donated by Dr. Luis Antonio Alonso Palomares and Dr. Isabel Salazar from the Virology and Immunovirology Laboratory of the "Escuela Nacional de Ciencias Biológicas del Instituto Politécnico Nacional, Mexico", ENCB-IPN, Mexico City.

We wish to acknowledge Juan Enrique del Águila Romero, Executive Director of the Red de Salud de Alto Amazonas, for his logistical support in the field work, and Dr. Vilma R. Béjar Castillo, Director of Instituto de Medicina Tropical "DAC", for her support of the project. Additionally, thank you to Kike Sinarahua Ishuiza, Estefany Janneth Garay Vela and Antolín Saldaño from Salud Ambiental de la Red de Salud de Alto Amazonas for their contribution to field work.

## Author Contributions

**Conceptualization:** Alvaro Aguilar-Setién, Jan Felix Drexler.

**Data curation:** Alvaro Aguilar-Setién.

**Formal analysis:** Alvaro Aguilar-Setién, Mónica Salas-Rojas, Guillermo Gálvez-Romero, Cenia Almazán-Marín, Andrés Moreira-Soto, Jorge Alfonso-Toledo, Daniel Felipe Barrantes Murillo, Alejandro Alfaro-Alarcón, Osvaldo López-Díaz, Jan Felix Drexler.

**Funding acquisition:** Alvaro Aguilar-Setién, Jan Felix Drexler.

**Investigation:** Alvaro Aguilar-Setién, Mónica Salas-Rojas, Guillermo Gálvez-Romero, Cenia Almazán-Marín, Andrés Moreira-Soto, Jorge Alfonso-Toledo, Cirani Obregón-Morales, Martha García-Flores, Anahí García-Baltazar, Daniel Felipe Barrantes Murillo, Dominique Pontier, Ondine Filippi-Codaccioni, Jean-Baptiste Pons, Jeanne Duhayer.

**Methodology:** Mónica Salas-Rojas, Guillermo Gálvez-Romero, Cenia Almazán-Marín, Andrés Moreira-Soto, Jorge Alfonso-Toledo, Cirani Obregón-Morales, Martha García-Flores, Anahí García-Baltazar, Jordi Serra-Cobo, Marc López-Roig, Nora Reyes-Puma, Marta Piche-Ovares, Mario Romero-Vega, Claudio Soto-Garita, Alejandro Alfaro-Alarcón, Eugenia Corrales-Aguilar, Osvaldo López-Díaz, Dominique Pontier, Ondine Filippi-Codaccioni, Jean-Baptiste Pons, Jeanne Duhayer.

**Project administration:** Alvaro Aguilar-Setién, Jan Felix Drexler.

**Supervision:** Jan Felix Drexler.

**Visualization:** Mónica Salas-Rojas, Cenia Almazán-Marín, Martha García-Flores.

**Writing – original draft:** Alvaro Aguilar-Setién, Guillermo Gálvez-Romero, Cenia Almazán-Marín, Andrés Moreira-Soto.

**Writing – review & editing:** Alvaro Aguilar-Setién, Andrés Moreira-Soto, Jordi Serra-Cobo, Daniel Felipe Barrantes Murillo, Osvaldo López-Díaz, Jan Felix Drexler.

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
