## [Decision Letter · Decision Letter 0]

5 Jun 2022

Dear Dr. Aguilar-Setien,

Thank you very much for submitting your manuscript "Experimental infection of Artibeus lituratus bats and no detection of Zika virus in neotropical bats from French Guyana, Peru, and Costa Rica, suggest a limited role of bats in Zika transmission." for consideration at PLOS Neglected Tropical Diseases. As with all papers reviewed by the journal, your manuscript was reviewed by members of the editorial board and by several independent reviewers. In light of the reviews (below this email), we would like to invite the resubmission of a significantly-revised version that takes into account the reviewers' comments. 

We cannot make any decision about publication until we have seen the revised manuscript and your response to the reviewers' comments. Your revised manuscript is also likely to be sent to reviewers for further evaluation.

Sincerely,

Richard A. Bowen

Associate Editor

Scott Weaver

Deputy Editor

Reviewer's Responses to Questions

**Key Review Criteria Required for Acceptance?**

**Methods**

-Are the objectives of the study clearly articulated with a clear testable hypothesis stated?

-Is the study design appropriate to address the stated objectives?

-Is the population clearly described and appropriate for the hypothesis being tested?

-Is the sample size sufficient to ensure adequate power to address the hypothesis being tested?

-Were correct statistical analysis used to support conclusions?

-Are there concerns about ethical or regulatory requirements being met?

Reviewer #1: It is crucial to establish whether the wild bats were captured in areas and times when ZIKV was actually circulating. Otherwise we cannot assume exposure and the field data have unclear value. Ideally these historical data should be provided at a sub-national level (ie, when was ZIKV first detected in the province/district where bats were sampled). We need to know how many bats were sampled in areas of ongoing ZIKV circulation. Having bats from before ZIKV introduction has value as a control, but the power to detect infection relies on the post-ZIKV samples. 

Unclear whether serology was used to to rule out prior exposures in the bats used for experimental infections. The text doesn’t mention it, but table 1 does. Be explicit.

Minor points:

Lacking information on how bats were housed for the captive study (e.g., number of bats in cages, whether they were separated by experimental group; when they were brought in from the wild etc).

P15. For field studies, more detail is needed on sample processing and storage (method of blood collection, buffers, cold chain etc). 

P15. For field studies, much of the detail on sampling effort is currently in the results, but would be better in methods. 

P16. Statement on bat fever (or lack thereof) from other viruses should be re-written or removed. Not sure what extra context this adds. 

P22/23. Missing a national authorization for bat capture and sampling in Peru.

Reviewer #2: While it is inferred in the introduction that this study aims to fill in a gap of knowledge (actual susceptibility of wild bats to ZIKV), it would be beneficial to clearly make this statement. Statistical analysis is lacking. Ethical/regulatory requirements for the handling/testing on bats are addressed. Below are some recommended edits for the methods section:

1) In Table 1A, it is shown that blood samples were taken at -1 day post-infection for ZIKV serology. What were the serological status (negative/positive, titer, etc) for the captured bats prior to the experimental inoculation? Please include this information under "Capture of bats for infection experiments."

2) Under "Virus stock," please fix C636 to C6/36.

3) Under "Experimental infection," please define the first usage of "WBC" for the readers.

4) Under "Histological studies," please include information about the histopathological analysis. For example, were the slides analyzed by veterinary pathologist? Was it a blinded analysis? Lesion scoring details?

5) Under "Plaque reduction neutralization test," what is the viral titer used? (i.e. "...serum dilution plus 35uL of ZIKV..."). Additionally, what is the reasoning behind using a different ZIKV strain for PRNT instead of using the same strain that was used for the experimental inoculation?

6) Under "A sampling of bats in different countries of Latin America," it states that geographic coordinates are included in Table S1. This information is not present in the table. Also, please include some information on how the blood samples were temporarily stored and transferred prior to analysis.

Reviewer #3: No major analyses or experiment needed.

Following minor revisions are recommended.

1. RT-PCR Assay name or primers names should be specified as the reference publication(29) includes several assays (PDF page 15). 

2. sampling sites and geographic coordinates are not included in the supplementary Table S1 as mentioned in the description (PDF page 15).

**Results**

-Does the analysis presented match the analysis plan?

-Are the results clearly and completely presented?

-Are the figures (Tables, Images) of sufficient quality for clarity?

Reviewer #1: The results largely concentrate on the physiological effects of infection. This is not my expertise, so I can't comment much on the quality/appropriateness. 

It would be helpful for future studies to provide raw data at the individual bat level as a supplementary table, epecially for the experimental infections.

Reviewer #2: Below are some recommended edits for the results section:

1) Table 1: It may be more appropriate to refer to +mild and ++moderate instead of "slight" and "medium." Please also include the legend for (-). Please also fix the superscript presentation of the RNA titers. In Panel B, the lines appear very slanted. Please align the lines and also add an additional vertical line between MO07 and MO11 to separate group 4 and 5.

2) Please be consistent in your animal IDs (i.e. M2 vs MO02, M9 vs MO09, etc).

3) Under "Complete blood count," please rewrite the section starting with "The most relevant results.." It is difficult to understand the main message based on the text and given table. 

4) Table 2: Is there a reason why values from only certain animals are presented in the table? It is difficult to interpret whether the increase or decrease is normal for that particular animal or not since their baseline (day -1 or 0 data) are not provided for all of the animals. The complete CBC data for each animal/each infection day could be provided as a supplementary table. Key significant/interesting findings could be presented as a in-text table or figure that must be complete and understandable on its own.

5) Under "Histopathology," please include in text what was observed in the control bats (male and female reproductive tracts, patagium, and spleen). 

6) Figure 2: Figure 2 controlled and Figure 2 infected could be combined into Figure 2 A and B. 

7) Under "Limited detection of viral RNA in urine," it is stated that "all other animals and TISSUES tested negative." Does this mean that all organ tissues collected had no ZIKV RNA detection? Does this also include the oral swabs? Please clarify in the text. 

8) Under "seroneutralization test," please include the serological status of the animals prior to inoculation. Also, please define the first usage of "(HIT)."

Reviewer #3: Following minor revisions are recommended.

1. Genders of MO04 and MO06 in Table 01 are contradictory to that of the methodology description (PDF Page 12, 3rd line from the bottom)

2.The sentence 'M9 presented a marginal erythrocytosis on day 21 post-inoculation' has to be corrected as the M9 subject was euthanized on day 14(PDF Page 16).

3. The sentence 'Between 3 and 7 days after infection, a clear leukopenia with neutropenia was observed in the animals M4, M6, M1, and M8, with recovery after day 9 pi.' should also be corrected as the animals were euthanized on day 3 and day 7 respectively(PDF Page 16).

4. In Table 02, it is better to mention the CBC values of M7, M10 and M11 on day 0 for better comparison(PDF Page 19).

5. In results section and in author summary, 33 genera should be corrected as 34 and genus name 'Mycronycteris' should be corrected as 'Micronycteris' (PDF Page 10 and 20).

6. In supplementary Table S1, genus name 'Chiroderna' should be corrected as 'Chiroderma'

**Conclusions**

-Are the conclusions supported by the data presented?

-Are the limitations of analysis clearly described?

-Do the authors discuss how these data can be helpful to advance our understanding of the topic under study?

-Is public health relevance addressed?

Reviewer #1: Evidence of clinical disease but without qPCR detection of viral RNA is confusing and seems worth discussing explicitly.

Limitations around field studies need to be discussed, especially with regard to sampling date and location). 

Limitations around captive infections need to be discussed; for example, the small sample size used, the short duration of the experiment, and uncertainty over whether different routes/doses of innoculation would have led to different outcomes. Also since the study serially euthanized, possibly before virus shedding might have manifested, we can’t be confident that more bats would not have shed or that titers wouldn’t have been higher (though absence from other organs is consistent with the authors’ interpretation). 

The logic behind the overall conclusion (bats might not have an important role in ZIKV transmission) needs to be carefully laid out. At present, it seems like results were generally not very different from previous studies but the authors somewhat subjectively reach the opposite conclusion. I do not necessarily disagree with the authors, but they need to explain how the results support their conclusion, which is currently missing from the discussion. 

Minor points:

P21. Sentence on ZIKV vs DENV is slightly unclear can be re-worded. 

P21. Statement on Aedes is important to mention somewhere in the manuscript, but as far as I can see has nothing to do with bat reproduction, which is the topic of the paragraph.

Reviewer #2: Below are some recommended edits for the discussion section:

1) Please begin this section with a new sentence that highlights the overall study findings or its importance. 

2) Third paragraph could include some information of the susceptibility of sylvatic neotropical mosquitoes to ZIKV as more supporting evidence. (i.e. Fernandes et al. 2019 article on ZIKV vector competence in South America)

3) Please include a section discussing the limitations and strengths/weaknesses of the study. Expand on reasons why different or consistent results were observed compared to previous works. This information is somewhat found sprinkled in the results section but please add them into the discussion section.

Reviewer #3: Following minor revision is recommended.

1.Explain further how the total of 180 urine samples were obtained to increase the clarity of the sentence (PDF Page 21). eg. daily collection of urine samples from infected bats does not tally with 180 (2*3+2*7+3*14+4*21=189).

**Editorial and Data Presentation Modifications?**

Reviewer #1: Abstract:

“free-ranging” – this implies that bats were experimentally infected and released, which presumably is not the case. Please revise the wording. 

“titles” typo (should be titers) 

Please add the timeframe of sampling for the wild bat sampling effort.

Please mention the serological result in the abstract, not just the qPCR. Together these are compelling evidence.

Please include the exact bat species in the abstract rather than just “Artibeus bats”

Introduction

P11. “No active or transmissible…” the previous sentence mentions molecular detections in wild bats. Presumably the authors have a higher criterion for ruling out active infection, so would be helpful to elaborate there. 

P11. Earlier studies on ZIKV from the 50s-60s are called ‘anecdotal’. I’m not sure what this means, but as these seem to have been published studies, it is unfair to dismiss these efforts. Explain their limitations.

Reviewer #2: 1) For efficient review, please present the text in double spaced format with page numbers and line numbers added.

2) Please review the entire manuscript. There are many grammatical or punctuation errors (ex. missing or extra periods, adding random hyphens, etc.) and long run-on sentences that need to be addressed. 

3) Please increase the quality of the histological images to better appreciate the lesions observed.

Reviewer #3: Following minor revision is recommended.

1.Labelling of the animals should be consistent throughout the manuscript eg. MO06 or M6

**Summary and General Comments**

Reviewer #1: Thre is a lot of interest in the role of bats as viral reservoirs. This study presents a combination of field and experimental infections studies aimed at understanding the role of neotropical bats in the transmission of ZIKV. In going beyond detection, this study provides important nuance and detail. As such the manuscript contains valuable data and would have been a considerable effort. The authors find relatively little evidence of productive ZIKV replication in experimentally infected bats and no evidence of infection in wild bats. They conclude that bats are unlikely to be significant for ZIKV transmission. As always, there are some limitations to the work that was done, particularly related to the constrained sample size and duration of the captive bat study and possibly related to the timing of field studies with respect to Zika circulation. In light of my comments elsewhere, the authors may need to either tone down their conclusions or explain why alternative explanations of the data are unlikely.

Reviewer #2: The authors presented an interesting study in which they investigated the actual susceptibility/infection of wild bats found in Latin America to ZIKV infection. It required intense collaborative efforts to collect the samples required for this study. Please address the recommended edits to improve the quality of the manuscript so that the significance of the study can be appropriately highlighted. Major edits in terms of grammar, punctuations, clarity in figures/tables, and missing important information of the study needs to be addressed to improve the quality of the manuscript.

Reviewer #3: This research is based on the experimental infection of Zika virus in 9 wild caught Artibeus lituratus bats from Mexico. Researchers also discuss the results from PCR viral screening of blood samples from 2056 bats belongs to 34 genera caught from French Guyana, Peru and Costa Rica from 2010 to 2012 and 2015 to 2019.

Publication is very impressive. I appreciate the authors’ effort on generating these valuable results. However, I would like to recommend copy editing to improve the manuscript further. Title should also be rearranged to clearly depict the content of the article.

PLOS authors have the option to publish the peer review history of their article (what does this mean?). If published, this will include your full peer review and any attached files.

Reviewer #1: No

Reviewer #2: No

Reviewer #3: No

Figure Files:

Data Requirements:

Please note that, as a condition of publication, PLOS' data policy requires that you make available all data used to draw the conclusions outlined in your manuscript. Data must be deposited in an appropriate repository, included within the body of the manuscript, or uploaded as supporting information. This includes all numerical values that were used to generate graphs, histograms etc.. For an example see here: http://www.plosbiology.org/article/info:doi%2F10.1371%2Fjournal.pbio.1001908#s5.
---

## [Editor Report · Decision Letter 1]

1 Oct 2022

Dear Dr. Aguilar-Setien,

Thank you very much for submitting your manuscript "Experimental infection of Artibeus lituratus bats and no detection of Zika virus in neotropical bats from French Guiana, Peru, and Costa Rica, suggests a limited role of bats in Zika transmission." for consideration at PLOS Neglected Tropical Diseases. As with all papers reviewed by the journal, your manuscript was reviewed by members of the editorial board and by several independent reviewers. In light of the reviews (below this email), we would like to invite the resubmission of a significantly-revised version that takes into account the reviewers' comments. 

Your manuscript has been carefully reviewed by experts in the field and all felt that this work provides a potentially valuable addition to our understanding of the role of bats in zoonotic transmission of arboviruses. Having said that, the reviewers indicated a rather large number of issues that need to be addressed to improve the clarity of the manuscript, and to more fully describe the techniques that were used and how certain conclusions were reached. Additionally, there was a general consensus that the manuscript is in need of careful proofreading to resolve problems with punctuation, grammar, spelling and inconsistencies in things like animal numbering.

Please evaluate the reviewer comments carefully, provided answers and clarifications to the points reviewers have raised and submit a revised version of the manuscript at your earliest convenience. We look forward to seeing a new version of this interesting manuscript.

We cannot make any decision about publication until we have seen the revised manuscript and your response to the reviewers' comments. Your revised manuscript is also likely to be sent to reviewers for further evaluation.

Sincerely,

Richard A. Bowen

Academic Editor

Scott Weaver

Section Editor

Your manuscript has been carefully reviewed by experts in the field and all felt that this work provides a potentially valuable addition to our understanding of the role of bats in zoonotic transmission of arboviruses. Having said that, the reviewers indicated a rather large number of issues that need to be addressed to improve the clarity of the manuscript and to a more fully describe the techniques that were used and how certain conclusions were reached. Additionally, there was a general consensus that the manuscript is in need of careful proofreading to resolve problems with punctuation, grammer, spelling and inconsistencies in things like animal numbering.

Please evaluate the reviewer comments carefully, provided answers and clarifications to the points reviewers have raised and submit a revised version of the manuscript at your earliest convenience. We look forward to seeing a new version of this interesting manuscript.

Figure Files:

Data Requirements:

Please note that, as a condition of publication, PLOS' data policy requires that you make available all data used to draw the conclusions outlined in your manuscript. Data must be deposited in an appropriate repository, included within the body of the manuscript, or uploaded as supporting information. This includes all numerical values that were used to generate graphs, histograms etc.. For an example see here: http://www.plosbiology.org/article/info:doi%2F10.1371%2Fjournal.pbio.1001908#s5.
---

## [Decision Letter · Decision Letter 2]

29 Mar 2023

Dear Dr. Aguilar-Setien,

Thank you very much for submitting your manuscript "Experimental infection of Artibeus lituratus bats and no detection of Zika virus in neotropical bats from French Guiana, Peru, and Costa Rica, suggests a limited role of bats in Zika transmission." for consideration at PLOS Neglected Tropical Diseases. As with all papers reviewed by the journal, your manuscript was reviewed by members of the editorial board and by several independent reviewers. The reviewers appreciated the attention to an important topic. Based on the reviews, we are likely to accept this manuscript for publication, providing that you modify the manuscript according to the review recommendations. 

This manuscript has been revised twice and it has been improved considerably. However reviewers still have some comments that should be addressed. One thing that definitely needs to be changed is the format of Table 2, which currently is very difficult to read. Please do 2 things with table 2: 1) change the font to the same (smaller) font used in the text so that the bat numbers do not wrap to the next line and the text is not so crushed together; alternatively, change the table to landscape mode so that it is easier to read, and 2) indicate in Table 2 and Figure 1 that bats 11 and 12 are the controls; you could do this by putting a * next to their numbers and indicating they are controls in the Table footer or Figure legend. One final comment: you indicate that some of the lesions observed by histopathology could be due to other pathogens or parasites since these animals were wild caught. It would be worth emphasizing this in the section beginning on line 366 where you describe the neurologic lesions in bat m008 - finding a neurologic lesion in one bat is far from convincing evidence that Zika virus was responsible. 

We look forward to reviewing a final revision ofd this manuscript.

Sincerely,

Richard A. Bowen

Academic Editor

Scott Weaver

Section Editor

This manuscript has been revised twice and it has been improved considerably. However reviewers still have some comments that should be addressed. One thing that definitely needs to be changed is the format of Table 2, which currently is indeed very difficult to read. Please do 2 things with table 2: 1) change the font to the same (smaller) font used in the text so that the bat numbers do not wrap to the next line and the text is not so crushed together; alternatively, change the table to landscape mode so that it is easier to read, and 2) indicate in Table 2 and Figure 1 that bats 11 and 12 are the controls; you could do this by putting a * next to their numbers and indicating they are controls in the Table footer or Figure legend. One final comment: you indicate that some of the lesions observed by histopathology could be due to other pathogens or parasites since these animals were wild caught. It would be worth emphasizing this in the section beginning on line 366 where you describe the neurologic lesions in bat m008 - finding a neurologic lesion in one bat is far from convincing evidence that Zika virus was responsible. 

We look forward to reviewing a final revision ofd this manuscript.

Reviewer's Responses to Questions

**Key Review Criteria Required for Acceptance?**

**Methods**

-Are the objectives of the study clearly articulated with a clear testable hypothesis stated?

-Is the study design appropriate to address the stated objectives?

-Is the population clearly described and appropriate for the hypothesis being tested?

-Is the sample size sufficient to ensure adequate power to address the hypothesis being tested?

-Were correct statistical analysis used to support conclusions?

-Are there concerns about ethical or regulatory requirements being met?

Reviewer #1: Regarding the Peruvian samples, my earlier comment related to the national permits usually required for collection of samples from wildlife (SERFOR), not the animal ethics approval from a local University. Perhaps there was some exception here, but I will leave this to the Peruvian co-author(s) to consider.

Reviewer #2: The message behind the study is very interesting but it is very hard to follow through the provided data tables and figures. For example, it's confusing to read based on how the tables were organized and very difficult to quickly obtain the significant finding(s). The presentation of the data must be improved or changed for the better so that the main contents of the manuscript (results and discussion) can be better evaluated by the reviewers. Below are some highly recommended suggestions:

1. Methods: Please explain why the same virus used for the inoculum is not used for the PRNT. Please include this discussion in the manuscript.

2. Methods, Fieldwork paragraph: Maybe providing the Zika cases/outbreaks in Peru, Costa Rica and French Guiana would be easier to follow if they were in a bulleted list. 

3. Methods, line 264: What euthanasia drug was used IM for these field-collected bats? Potential hematologic effect to viral RNA isolation and detection? Please include this discussion in the manuscript.

4. Table 1: Apply alternating color or borders to make it easier for the reader to follow the numbers to the species.

5. Results, line 284: Change "though" to "through."

6. Results, line 286: Were there any pathogen detection panels used to identify potential subclinical/latent infections that may not cause clinical disease but could potentially cause hematologic alterations or microscopic lesions? Please include this discussion in the manuscript. 

7. Figure S1: Please verify this is the correct data figure. Data points do not match the study. For example, for the 3DPI group (2 bats; the black filled in dots), there are for some reason three data points in day 3 and then also at day 8, 9, and 11? It is stated in the methods section that 3DPI group bats were euthanized at 3 DPI. How are there data points for this group at day 8, 9, and 11 post inoculation? Please explain.

8. Table S1: Please include the row label for this table. It is unclear what the rows mean.

9. Table 2: Please change this table into landscape orientation layout to fit the column labels more appropriately. Also, please include column borders to separate the DPIs. This table is very difficult to read and to appreciate the findings. 

10. Histopathology: Highly recommended to include immunohistochemical staining of the tissues to further support that the microscopic lesions observed are due to the Zika virus infection and not from other adventitious or subclinical infection. 

11. Line 396: Where is this Table 1A?

**Results**

-Does the analysis presented match the analysis plan?

-Are the results clearly and completely presented?

-Are the figures (Tables, Images) of sufficient quality for clarity?

Reviewer #1: No comments

Reviewer #2: (No Response)

**Conclusions**

-Are the conclusions supported by the data presented?

-Are the limitations of analysis clearly described?

-Do the authors discuss how these data can be helpful to advance our understanding of the topic under study?

-Is public health relevance addressed?

Reviewer #1: The first sentence of the discussion implies that no ZIKV antibodies were found in 2056 tested bats. I don’t see these data mentioned in the results in the section on the free living bats. Similarly, the methods for the field study mentions only PCR, not serology. Please revise this sentence if no serology was done on the free living bats, or revise the methods and results if serology was done on these samples and no positives were found. 

I am relieved to see that many samples from free-living bats were collected post-Zika, but I still think some extra caution is needed here since we don't know what the local incidence of Zika infection was at the time of bat sampling, whether the specific areas tested had evidence of Zika circulation at all, or what % of bats infected would indicate a substantial role in Zika transmision. The overall sampling effort is large, but if were were to hypothesize for example that only 1 or 2 bat species are relevant, sample sizes suddently appear quite small. I urge the authors to add a sentence or two to the discussion acknowledging these limitations. This section should also include a statement on serum vs urine (currently in the results section) which is important to contextualize the negative results. It should also include some explanation for why serology was not done on the serum samples from free living bats, if that was indeed the case (see above).

Reviewer #2: (No Response)

**Editorial and Data Presentation Modifications?**

Reviewer #1: Table 2. Difficult to see which are the controls and to work out which columns correspond to each day post infection.

Reviewer #2: (No Response)

**Summary and General Comments**

Reviewer #1: The authors have done a nice job revising the manuscript. I'm glad to see that most samples from the free living bats were post-ZIKA arrival and the clarifications around that point improve the paper. However, I would still encourage the authors to more critically evaluate these data and what they can tell us about the role of bats for Zika transmission in the discussion section.

Reviewer #2: The message behind the study is very interesting but it is very hard to follow through the provided data tables and figures. For example, it's confusing to read based on how the tables were organized and very difficult to quickly obtain the significant finding(s). The presentation of the data must be improved or changed for the better so that the main contents of the manuscript (results and discussion) can be better evaluated by the reviewers.

PLOS authors have the option to publish the peer review history of their article (what does this mean?). If published, this will include your full peer review and any attached files.

Reviewer #1: No

Reviewer #2: No

Figure Files:

Data Requirements:

Please note that, as a condition of publication, PLOS' data policy requires that you make available all data used to draw the conclusions outlined in your manuscript. Data must be deposited in an appropriate repository, included within the body of the manuscript, or uploaded as supporting information. This includes all numerical values that were used to generate graphs, histograms etc.. For an example see here: http://www.plosbiology.org/article/info:doi%2F10.1371%2Fjournal.pbio.1001908#s5.

Reproducibility:

References

---

## [Editor Report · Decision Letter 3]

21 Jun 2023

Dear Dr. Aguilar-Setien,

We are pleased to inform you that your manuscript 'Experimental infection of Artibeus lituratus bats and no detection of Zika virus in neotropical bats from French Guiana, Peru, and Costa Rica, suggests a limited role of bats in Zika transmission.' has been provisionally accepted for publication in PLOS Neglected Tropical Diseases.

Best regards,

Richard A. Bowen

Academic Editor

Scott Weaver

Section Editor

---

## [Editor Report · Acceptance letter]

19 Jul 2023

Dear Dr. Aguilar-Setién,

We are delighted to inform you that your manuscript, "Experimental infection of Artibeus lituratus bats and no detection of Zika virus in neotropical bats from French Guiana, Peru, and Costa Rica, suggests a limited role of bats in Zika transmission.," has been formally accepted for publication in PLOS Neglected Tropical Diseases.

Best regards,

Shaden Kamhawi

co-Editor-in-Chief

Paul Brindley

co-Editor-in-Chief
